# Effects of Bamboo Leaf Fiber Content on Cushion Performance and Biodegradability of Natural Rubber Latex Foam Composites

**DOI:** 10.3390/polym15030654

**Published:** 2023-01-27

**Authors:** Keavalin Jitkokkruad, Kasama Jarukumjorn, Chaiwat Raksakulpiwat, Saowapa Chaiwong, Jutarat Rattanakaran, Tatiya Trongsatitkul

**Affiliations:** 1School of Polymer Engineering, Institute of Engineering, Suranaree University of Technology, Nakhon Ratchasima 30000, Thailand; 2Center of Excellence on Petrochemical and Materials Technology, Chulalongkorn University, Bangkok 10330, Thailand; 3Research Center for Biocomposite Materials for Medical Industry and Agricultural and Food Industry, Suranaree University of Technology, Nakhon Ratchasima 30000, Thailand; 4School of Agro-Industry, Mae Fah Luang University, Chiang Rai 57100, Thailand; 5Integrated AgriTech Ecosystem Research Group, Mae Fah Luang University, Chiang Rai 57100, Thailand

**Keywords:** natural rubber latex foam, bamboo leaf fiber, foam composite, foam net, Dunlop process, microwave-assisted vulcanization, cushion coefficient, biodegradability

## Abstract

Bamboo leaf fiber (BLF) was incorporated into an eco-friendly foam cushion made from natural rubber latex (NRL) to enhance the biodegradation rate. The objective of this work was to investigate the effects of BLF content on the foam structure, mechanical properties, cushion performance, and biodegradability. The NRL foam cushion nets with and without BLF were prepared using the Dunlop method along with microwave-assisted vulcanization. BLF (90–106 µm in length) at various loadings (0.00, 2.50, 5.00, 7.50, and 10.00 phr) were introduced to the latex compounds before gelling and vulcanizing steps. Scanning electron microscopy (SEM) showed that the BLF in a NRL foam caused an increase in cell size and a decrease in the number of cells. The changes in the cell structure and number of cells resulted in increases in the bulk density, hardness, compression set, compressive strength, and cushion coefficient. A soil burial test of 24 weeks revealed faster weight loss of 1.8 times when the BLF content was 10.00 phr as compared to the NRL foam without BLF. The findings of this work suggest the possibility of developing an eco-friendly cushion with a faster degradation rate while maintaining cushion performance, which could be a better alternative for sustainable packaging in the future.

## 1. Introduction

Plastic waste is one of the most problematic among various materials. The beneficial qualities such as its light weight, durability, inertness, ease of the process, and versatility of plastics have served humans well for several decades. On the other hand, these qualities also lead to an enormous amount of plastic waste accumulation due to the high consumption volume and poor decomposability/degradability. The UN environment program (UNEP) anticipates that plastic will increase by 1100 million tons in 2050 from 400 million tons in the present year. Because most plastics are fossil-based materials, the widespread use of these materials adds other concerns to environmental issues, such as the exhaustion of natural resources and global warming [1]. Several key aspects must be considered to effectively address these problems, including product design and material selection to enhance recyclability [2], strict regulation enforcement [3], and proper education of the public [4] are examples of how to remedy the situation sustainably.

Because plastics are versatile materials, they are used in several industries, such as automotive, construction, home appliances, electronics, and packaging [5]. The proper management of plastic product waste requires an understanding of their natures, including the type of materials used, how they are made, how they are used, service life, and post-consumer disposal. Among all plastic products, approximately 36% is used in packaging. Single-use items such as food and drink containers are made up of the most share of waste. Although rigid plastic packaging, such as bottles, boxes, trays, and cups, has been successfully recycled [6], 85% of plastic packaging waste is discarded as uncontrolled waste or in landfills [7]. These problematic wastes are mainly flexible packaging, such as plastic films and bags, as well as cushioning materials. These types of packaging are difficult to collect as they are very thin or/and low in bulk density. They are also difficult to separate (multi-layer film) and once contaminated, they are also difficult to clean. Recycling these plastics waste is often logistically and economically unfeasible. For example, cushion foam nets for fresh produce protect the goods during transportation. Once the produce arrives customers, the cushion is discarded even though it could have been reused or recycled. The cost to retrieve, collect, and transport back to the packaging location is cumbersome and requires combined efforts from several parties.

Plastics used to fabricate cushion foam nets for fresh produce are mainly made from expanded polystyrene foam (EPS) and expanded polyethylene foam (EPE). With the foam structure, they are exceptionally light and able to absorb mechanical shock, vibration, and impact forces that occur during transportation. Fresh fruit covered with proper cushion materials are therefore protected from mechanical damage. Together with good cold-chain management, the fruit’s appearance, quality, and nutrients can be preserved [8]. Cushioning is thus a crucial part of a packaging system. However, to reduce the negative impact of the synthetic foams [9,10], many countries have placed restrictions on their use. This includes laws imposed by local and state governments on single-use packaging at the end of its life cycle. Consequently, an alternative cushion that is eco-friendly and possesses a good cushion performance is in great demand.

To substitute the current nondegradable polystyrene and polyethylene packaging materials, researchers have focused on using biodegradable polymers, such as polylactic acid (PLA), poly(lactic-co-glycolic acid) (PLGA), polybutylene adipate terephthalate, (PBAT), and poly(3-hydroxybutyrate-co-3-hydroxyvalerate) (PHBV) [11]. However, these polymers are substantially more expensive compared to conventional petroleum-based polymers [12], such as high-density polyethylene (HDPE), low-density polyethylene (LDPE), and polypropylene (PP), etc. The lack of processability and, even less so, their cushioning performance information lead to the unsuccessful use of these biodegradable polymers. Blends of biodegradable polymers and conventional polymers, such as HDPE/PLA [13], PLA/PP [14] and PLA/LDPE [15], etc., have also been investigated. Their practicality and implementation, as well as environmental impacts, are yet to be evaluated.

Another interesting alternative for cushioning materials is natural rubber (NR). NR is harvested from the rubber tree (*Hevea brasilensis*) in latex [16,17]. NR is, therefore, bio-based material that comes from a renewable resource and is inherently biodegradable [18]. In fact, a NR product may undergo degradation via one or a combination of several degradation modes, including mechanical degradation, oxidation, photodegradation, thermal degradation, and/or biodegradation [19]. Previous works have been done to prevent the degradation of a NR product to extend its service life. Now degradation of NR is appreciatively viewed as a solution for environmental pollution. Foam products of NR have been used as mattresses, pillows [16,20], cushions [17], padding foam [21], etc. The NRL foam possesses unique properties such as lightweight, good thermal insulation, good sound absorption, excellent elasticity, good cushioning performance, and biodegradability [20,22,23]. Our research team is interested in using NRL foam as cushion material for fresh produce.

Generally, the NRL foam can be fabricated using a Dunlop or Talalay process. The Dunlop process has been widely used as it is a simpler process with a lower production cost and is more energy efficient than those of the Talalay process [24]. The Dunlop process steps include incorporation of the air and some chemicals into the latex via a mechanical method to create and stabilize the foam structure before vulcanization. Heating in the vulcanization step may use either a conventional hot air oven and a microwave oven [25]. The recent trend of replacing conventional heating with microwave irradiation is due to the advantages of the later process. For the microwave heating, the heat generated in the product results from the conversion of electromagnetic energy to thermal energy [26]. Because of the direct interaction between the microwave and the heated article, energy transfer occurs more efficiently. Thus, microwave heating provides a significantly faster heating rate, resulting in shorter vulcanization time, higher production rate, and less time and energy consumption compared to conventional heating [25,27]. We have successfully optimized the microwave irradiation condition for making the NRL foam [28]. The study showed the promising and important finding that the use of microwave heating could reduce vulcanization time by 15-fold (from 90 min [24] to 6 min [28]). Using microwave irradiation for NRL foam fabrication is, therefore, fundamentally making the foam product more eco-friendly.

To further improve the eco-friendly attributes of NRL foam, enhancing its biodegradation rate is considered. Even though NR is fully degraded, the degradation time of rubber products could be up to several decades. The relatively long degradation time may lead to environmental pollution as landfill sites become too limited [19,29]. A biodegradable filler may be added to the NRL to reduce such issues. Natural fillers, such as rice husk [30,31], fiber of banana, coir, bagasse [32], oil palm fiber [33], sisal [34], bamboo [35], and kenaf [36,37] have been added to polymer matrixes. Several research groups have reported that the addition of natural fibers contributed to the enhancement of biodegradability for the polymer products [30].

Bamboo is one of the most common plants available in Asian countries. It is one of the fastest-growing plants, is widely accessible, considered a sustainable resource [38,39], and is strong and lightweight [40]. Bamboo leaf is the by-product of growing bamboo trees. It is high in fiber, protein, and silica content and can be used for bamboo tea, bamboo beer, livestock feed, medicinal aids, aromatherapy, and essential oils [41]. Research and development have found bamboo leaves to be a good carbon source. They can be made into aerogel, silica nanoparticles, adsorbents, and composites. Adding BLF into NRL foam should positively affect the environment, society, and the economy [42]. However, limited information is available regarding how the BLF affects the cushion performance of NRL foam, especially for fresh produce packaging applications.

Therefore, the aim of this study is to develop eco-friendly cushioning packaging from NRL foam composites. The main focus is to investigate the effect of BLF content on mechanical properties, cushion performance, and biodegradability. The NRL foam composites containing different BLF contents (0.00, 2.50, 5.00, 7.50, and 10.00 phr) were prepared into a similar shape to the commercial foam net cushion using the Dunlop process and vulcanization via microwave heating. The NRL foam composites were compared to a commercial EPE cushion net in terms of foam density, hardness, morphology, cushion coefficient (C), and biodegradability. Additionally, the composites’ mold shrinkage, compression set, number of cells per unit volume, and average cell size were studied. A soil burial test of the samples was carried out for a duration of 24 weeks (6 months). The degradation of the NRL foam composites were analyzed via weight loss, Fourier transform infrared (FTIR) analysis, foam appearance, morphology, and compressive properties.

## 2. Materials and Methods

Chemical ingredients and their contents used in the formulation of the NRL foam composites are listed in Table 1. The ingredients were supplied by Chemical and Materials Co., Ltd., Bangkok, Thailand. Dried bamboo leaf was purchased from local farmers in Nakhon Ratchasima, Thailand. The BLF preparation is described below.

### 2.1. Preparation of Bamboo Leaf Fiber

Dried bamboo leaf was ground in a fine wood crusher machine (WSC-20, CT, Samut Prakan, Thailand) for 1 h. The ground BLF was then sieved using a vibratory sieve shaker (Analysette 3 Pro, Fritsch, Idar-Oberstein, Germany). Fibers with lengths in the range of 90–106 μm were obtained. The appearance of the BLF fiber is shown in Figure 1a. The BLF appeared as greenish particles, and SEM micrograph in Figure 1b reveals the irregular shape of the BLF particles. To use the BLF in the formular, the prepared fibers were soaked in deionized (DI) water with a 1:1 ratio of fiber to water.

### 2.2. Preparation of Rapid Silicone Mold

In this work, 3D printing was used to prepare a master with the desired shape of a cushion net, which was then used for making a negative cavity silicone mold. 3D printing has many benefits. The technique gives flexibility in designing molds and products. The process is fast and cost-effective compared to that of a conventional metal mold. The resulting silicone mold is also lightweight with good heat resistance and, most importantly, is suitable to be used in a microwave oven. First, the SolidWorks program (Dassault Systèmes, Vélizy-Villacoublay, France) was used to create a 3D model of the cushion net (Figure 2a). Then a CAD file was generated and converted to G-code using the PrusaSlicer program (Prusa Research, Prague, Czech Republic) (Figure 2b) before printing (Figure 2c). PLA filament (Prusament Prusa Galaxy Black, Prusa Research, Prague, Czech Republic) was used to print the inverted mold via a 3D printer (Original Prusa i3 MK3S, Prusa Research, Prague, Czech Republic). Silicone rubber and hardener, supplied by Infinite Crafts Co., Ltd., Bangkok, Thailand were mixed before pouring into the inverted mold. The silicone mold was left at room temperature until fully set. The working mold was obtained after the inverted mold was removed. The inverted and silicone molds are shown in Figure 2d and Figure 2e, respectively.

### 2.3. Fabrication of NRL Foam Composites via the Dunlop Process Together with Microwave-Assisted Vulcanization Technique

The NRL foam composites were prepared using the Dunlop process to generate a foam structure before curing via microwave irradiation. The process used here is similar to that reported previously [28] with the additional step of adding BLF. In short, the NRL was first stirred using a mechanical stirrer (Eurostar 20 digital, IKA Works, Wilmington, NC, USA). The stirring speed of the first step was 300 rpm for 30 min to reduce the ammonia content in the latex. Secondly, chemical ingredients required for the foaming process, i.e., K-oleate, Sulfur, ZDEC, ZMBT, and Wingstay L were incorporated, and the latex mixture was whipped at a speed of 1250 rpm for 6 min. Thirdly, a soaked bamboo leaf fiber was added to the whipped latex mixture at a speed of 500 rpm for 2 min. The BLF to DI water ratio was 1:1. Finally, ZnO, DPG, and SSF were added to the whipped latex for foam gelation at a speed of 700 rpm for 1 min. The gelled foam was poured into a silicone mold and set at room temperature for 1 h. The foams were vulcanized using a commercially available domestic microwave oven (MS23K3513AW/TC, Samsung, Kuala Lumpur, Malaysia). The optimum vulcanization condition was used, i.e., microwave irradiation power and time of 600 watts and 6 min, respectively [28]. Finally, the NRL foam product was washed with water and dried in a hot air oven at 70 °C for 4 h.

### 2.4. Characterization

#### 2.4.1. Bulk Density

The bulk density of NRL foam and its composites were calculated using the following Equation (1) [20]:(1)$$Bulk\;density=\frac{M}{V}$$
where: *M* is the mass of specimen (kg), *V* is the volume of the specimen (m^3^).

#### 2.4.2. Hardness

The hardness values of the NRL foam composites were determined using a durometer hardness (Durometer LX-OO Shore OO, X.F, Graigar, Guangdong, China) according to ASTM D2240 [43]. The hardness of the cushion foam net was measured on the sample surface. Five measurements on five different locations on each specimen were measured. The values reported were averaged from at least five replicates.

#### 2.4.3. Mold Shrinkage

The shrinkage of the NRL foam composites can be determined using the following Equation (2). The shrinkage is compared in three dimensions (width, length, and height). The reported values were averaged from at least five replicates.
(2)$$\mathrm{Shrinkage}\left(\%\right)=\frac{\left(X_{0}-X_{1}\right)}{X_{0}}\times 100$$
where $X_{0}$ are the dimensions of mold (mm), $X_{1}$ is the dimension of NRL foam composite after vulcanization (mm).

#### 2.4.4. Morphology

The cellular morphology of the cushion foams was analyzed using a scanning electron microscope (SEM) (AURIGA, Zeiss, Germany). Surface and cross-section of the foam samples were investigated. A foam sample was prepared from a small piece using a sharp blade. The surface and cross-section were sputter-coated with gold at 5.7 nm (Leica EM ACE600, Vienna, Austria). The SEM micrographs were taken at an accelerating voltage of 3 kV and working distance of 6.7–9.8 mm.

ImageJ image analyzer software was used to determine the foam’s cell size and cell count. For each cell, four lines were drawn across the cell, and the length of each line was measured. For each location or image, at least 100 cells were measured and recorded. Then, cell size distribution and average cell size were determined using a nonlinear curve fit (Gaussian) in the Origin program (OriginLab Corporation, Northampton, MA, USA). Additionally, the cell wall thickness of the cushion foams was measured using ImageJ software (NIH, Bethesda, MD, USA) and reported using average data from five locations of cell wall thickness.

The number of cells per unit volume (*N*) was evaluated by the averaged cell size data from ImageJ software and the foam density. Equation (3) was used to calculate the number of cells per unit volume of the NRL foam composite, as shown below [24]:(3)$$N=\left(\frac{6}{\pi d^{3}}\right)\left(\frac{\rho_{rubber}}{\rho_{foam}}-1\right)$$
where *N* is the number of cells per unit volume, *d* is the average cell diameter (cm), $\rho_{rubber}$ is a density of the solid rubber (1.09 g/cm^3^), and $\rho_{foam}$ is the density of the rubber foam (g/cm^3^) or the bulk density.

#### 2.4.5. Compression Test

A compression test of the cushion foam nets was performed using a universal testing machine (UTM) (Instron 5565, Norwood, MA, USA). The test was carried out using the load cell of 1 kN at a crosshead speed of 12 mm/min. The NRL composite foam net was cut into a rectangular shape with the dimensions of (width × length × thickness) 100 mm × 100 mm × 3 mm. The compression test was adapted from the static compression method for packaging buffer material (GB/T 8168-2008) to determine the cushion performance of cushion materials [44]. The relationship between stress and strain was obtained. The raw data of the compressive stress–strain curve was used to calculate the cushion coefficient. Moreover, 50% compressive strength was reported in the mechanical properties of the biodegradation study. The reported results were the averaged values from testing at least five replicates.

#### 2.4.6. Cushion Coefficient (*C*)

The cushion coefficient (*C*) is an ability of a cushion material to absorb the applied energy. The calculation was carried out using the data from the compression test mentioned earlier. The cushion coefficient equation is as follows [45]:(4)$$C=\frac{\sigma }{e}$$
where σ is the compressive stress (N/mm^2^), *e* is the energy absorption of the material (N·mm/mm^3^), which is estimated from the compressive stress–strain curve using Equation (5) [45]:(5)$$e=\int_{0}^{\varepsilon }\sigma \;d\varepsilon$$
where ε is the compressive strain (mm/mm).

#### 2.4.7. Compression Set

Compression set measurements were performed as per ASTM D1055 [46]. A cylindrical specimen with a diameter of 29 mm and a height of 19 mm was prepared using the same processing condition as the NRL composite foam net. Using a compression set apparatus, the specimen was compressed to 50% of its original thickness. The specimen was then placed in a hot air oven at 70 °C for 22 h. Afterward, the specimen was removed from the oven, and the compression set apparatus was removed from the specimen before being left to cool at room temperature for 30 min. The final thickness of the specimen was measured within 10 min after the cooling step. The compression set was calculated according to Equation (6) [46]:(6)$$C_{h}=\frac{\left(t_{0}-t_{1}\right)}{t_{0}}\times 100$$
where $C_{h}$ is the compression set, $t_{0}$ is the original thickness of the specimen (mm), $t_{1}$ is the thickness of the specimen after 30 min of removal from the compression set apparatus (mm).

#### 2.4.8. Biodegradation Study

The soil burial method was used to investigate the effect of BLF content in NRL foam composites on their biodegradability. Square specimens with the size of 50 mm × 50 mm × 3 mm (width × length × thickness) of each NRL composite foam net sample were placed in a box containing planting soil at the dept of 12–15 cm. The 100 L plastic box (500 mm × 770 mm × 43 mm) used was covered with a plastic lid, and small holes of 5 mm in diameter were uniformly drilled on all sides of the box for ventilation (see Figure 3). The soil used was planting soil from Suranaree farm, Suranaree University of Technology, Nakhon Ratchasima, Thailand. The plastic boxes filled with soil were placed outdoors beside equipment building 4 (F4) at Suranaree University of Technology, Nakhon Ratchasima, Thailand. The soil’s moisture content was maintained at 60–80% (water as needed). The commercial cushion EPE foam net (EPE-FN) was also buried in the soil with the same protocol and condition. The degradation study was carried out over a period of 24 weeks (May–October 2022). Every 4 weeks, five specimens of each sample were taken out from the soil and then washed with water and dried in a hot-air oven at 70 °C for 4 h. These specimens were tested for changes in the foam appearance, morphology, compression property (at 50% strain), and weight loss. The compression property and weight loss values were reported as averaged values calculated from at least three replicates. The weight loss formula [30] is given as Equation (7). An analytical balance with four digits (ML240, Mettler Toledo, Greifensee, Switzerland) was used for the weight loss determination.
(7)$$\mathrm{Weight}\;\mathrm{loss}\;\left(\%\right)=\frac{\left(W_{0}-W_{1}\right)}{W_{0}}\times 100$$
where $W_{0}$ is the sample weight before the test (g) and $W_{1}$ is the sample weight after the test (g).

#### 2.4.9. Fourier Transform Infrared (FTIR) Analysis

A FTIR spectrometer (Vertex 70-RamII, Bruker, Billerica, MA, USA) was used in attenuated total reflection (ATR) mode to study the change in the chemical structure of the samples after soil burial for 24 weeks. A small piece of the cushion foam sample was recorded at 400–4000 cm^−1^, and 64 scans were averaged at a resolution of 4 cm^−1^.

#### 2.4.10. Statistical Analysis

The SPSS software for Windows version 20 (SPSS Inc., Chicago, IL, USA) was implemented to conduct the statistical analysis of the cushion foam properties. Tukey’s HSD post-hoc test was used to compare the means of five replicates of bulk density, hardness, mold shrinkage, and compression set, and 400 measurements of average cell size for each cushion foam sample at the 0.05 significance level.

## 3. Results and Discussion

An eco-friendly cushion foam net was developed by the NRL. The aim of this development was to create an alternative cushion for the nonbiodegradable commercial EPE foam net used in fresh produce packaging. The NRL foam net was made from a sustainable resource using an energy-efficient process. Though natural rubber is generally considered inherently biodegradable, an increase in its biodegradation rate is desirable. The addition of BLF to the NRL foam net was mainly to enhance biodegradability. However, adding BLF to NRL may affect several other aspects, including processability, mechanical properties, appearance, and cushion performance. This work is focused on the investigation of the effects of the addition of BLF and its content into NRL foam cushion based on these aspects.

The NRL foam net was fabricated using the Dunlop process with microwave vulcanization. In the Dunlop process, the air was incorporated into the NRL and stabilized with chemicals. The appearance of the resulting NRL foam at this stage was similar to that of whipped cream, which was relatively delicate. Adding dried BLF directly to the compound caused the NRL foam to destabilize and collapse instantly. This could be attributed to the dried BLF absorbing significant amounts of moisture from the foam compound, destroying the delicate balance of the foamed rubber. Adding the moisture to the BLF, therefore, solved the problem and the NRL composite foams with different BLF contents were successfully fabricated.

The following sections are the results and discussion of the effects of the BLF addition into NRL foam based on their appearance and properties related to the fruit cushion packaging application.

### 3.1. Appearance and Design of the NRL Foam Cushion

The NRL foam cushion was made so that it could be tested and compared with a conventional EPE foam cushion. A flat net sheet was first made (see Figure 4a), and it was rolled to form a tube shape before fastening with rubber glue (see Figure 4b). The final shape of the NRL foam net (NRL-FN) cushion resembled that of the fruit foam net used commercially (EPE-FN) (Figure 5a). It should be noted that the flat net sheet of NRL-FN was comprised of one layer of square-cross-section filaments (see Figure 4d). This design was used to simplify the process of making the prototype cushion foam net. On the other hand, the commercial EPE-FN comprised round-cross-section filaments crossing over one another (see Figure 4c).

The NRL-FN with and without BLF are shown in Figure 5b–f. The NRL-FN without BLF was off-white with a slight yellow tint. Adding BLF to the NRL-FN resulted in a green color and its intensity increased with the BLF content. The green color was the natural color of BLF, as shown previously in Figure 1a.

#### 3.1.1. Mold Shrinkage

The shrinkage of cushion foam is an important parameter for cushion packaging design and fabrication. The shrinkage of NR products after vulcanization may have an impact on the final density and performance. The understanding of this shrinkage behavior can also be used to design the mold so the desired shape and dimension of the cushion can be obtained. The shrinkage of NRL-FN at different BLF contents is displayed in Figure 6. It could be generally seen that the height of all samples showed the highest shrinkage rates compared to the width and length. The height shrinkage further increased with the presence of BLF and with the increasing content.

On the other hand, the width and length slightly decreased with increasing BLF loading. The different shrinkages in height from the width and length were plausibly due to freedom of shrinkage from the top, the free surface of the molding foam during fabrication. Adding the higher-density ingredient of BLF into the foamed NR increased the weight that was pulled on the free surface of the molding NR foam. The side walls prohibited the changes in the width and length from molding the NR foam and prevented shrinkage in these directions.

The higher shrinkage rate observed when higher filler contents were incorporated into the foam was also reported by Ramli, R. et al. [47]. Moreover, the addition of filler was most likely to corrupt the foam’s cell structure [17]. Consequently, the density of the composite foam increased. The effect of BLF incorporation into the NRL-FN and its contents on the bulk density and cell structures are discussed in the following section.

#### 3.1.2. Bulk Density

In packaging, low bulk density material is preferred due to the lower fuel consumption and, thus, a lower cost of transportation. Table 2 shows the bulk density of EPE-FN compared to NRL-FN and NRL-FN-BLFs. It was clear that the density of EPE-FN commercial foam was greatly lower than that of the NRL-FN-BLFs (≥10 folds). This was expected as polyethylene (PE) possessed a lower density than NR. In addition, PE had a good melt strength, which was required to fabricate a foam structure with high porosity (large cell and thin wall). The SEM micrographs in the following section were in good agreement with the bulk density result.

The bulk density of NRL-FN was approximately 265 kg/m^3^. The density was higher than those values reported by Prasopdee, T. and W. Smitthipong (~65 kg/m^3^) [13] and Mahathaninwong, N. et al. (~161.6 kg/m^3^) [14]. Generally, the low density of NRL foam was achieved by the incorporation of the air into the NR latex or whipping in the Dunlop process [48]. Foaming time, foaming speed [28], and blowing agent content [22,49] were used to adjust the bulk density of the NRL foam. The differences in the density of the NR forms reported here as compared to others may be due to the different methods and/or equipment used during the aeration process. The addition of the BLF, which possessed higher density than the NRL foam, gave rise to the bulk density of NRL-FN-BLF composites. The presence of BLF also induced a higher degree of shrinkage (Figure 6), resulting in an increase in bulk density. A similar result was also found by Zhang, N. and H. Cao when chitin was used to enhance the antibacterial activity in NRL foam. The volume of the air in the NRL foam was decreased because chitin particles agglomerated and destroyed the original cellular structure of the foam cells, resulting in the collapse of the foam and thus increasing the foam bulk density [50].

#### 3.1.3. Morphology and Cell Structure Analysis

The foam surface and the foam cell characteristics of the EPE-FN and NRL-FN with and without BLF cushions were studied using SEM. The surfaces of the foam samples are shown in Figure 7. The EPE-FN surface shown in Figure 7a was dense and smooth without any holes or protrusions on the surface. On the other hand, the surface of the NRL-FN and NRL-FN-BLFs (Figure 7b–f) appeared to be dense and rough, with some holes and lumps present on the surface of some samples. The addition of BLF to the NRL-FN slightly caused the change in the surface roughness. The roughness increased slightly as a function of BLF content. No obvious fiber was found protruding from the surface. The roughness of the foam surface may implicate its use as cushioning for fresh produce. As the cushion foam surface is usually directly in contact with the fresh produce’s surface, the rough surface may cause bruising damage due to the rubbing that can occur during transportation. However, several other parameters can also contribute to accrued bruise damage on some fruit, including the foam’s hardness, cushion efficiency, fruit firmness, etc. The pack test of the target fruit using cushions with different BLF contents must be done to elucidate the effect of surface roughness of the NRL-FN on fruit surface bruising.

SEM micrographs of the cross-sectioned samples are shown in Figure 8. For EPE-FN (Figure 8a), the foam appeared to be a closed-cell structure with a very large cell size of 600–900 µm. The cell wall was relatively thin, with a thickness of 2–5 µm. The shape of the foam cell was a polygon with interconnecting thin walls similar to that of the honeycomb structure. A SEM micrograph of the NRL-FN foam structure in Figure 8b–f also revealed a more or less closed-cell structure of a spherical shape cell morphology with opens on some cell walls. Spherical foam cells were scattered throughout the sample. The spherical foam cells with different sizes were relatively small as compared to that of the EPE-FN. Cell size was analyzed using the ImageJ program and shown as a cell size distribution curve in Figure 9. The average cell size of the samples was calculated and shown in Table 2. The difference in cell size in a sample indicated that the air bubbles generated during the Dunlop process were nonuniform. This is a common result of a mechanical foaming technique, such as in the Dunlop process, where poor pore formation regulation is observed [19]. The cell wall was relatively thick, with a thickness in the range of 10–70 µm. The structures of the obtained NRL foams agreed with the high bulk density results discussed earlier. The lower bulk density of the NRL foam reported by [18,46] was due to the open cell structure of NRL they obtained. The polymeric foam can be made into either open or closed-cell foam, depending on the stabilizing agent used [18]. The results suggested that our NRL foam could be further optimized to obtain a lower bulk density to make it suitable for packaging applications. However, such a structure may need to prove its ability to withstand the incorporation of BLF.

The addition of BLF fiber into NRL-FN resulted in larger foam cells. As the BLF content increased, the cell size of the NRL-FN-BLF composites noticeably increased. On the other hand, the number of foam cells decreased as the BLF content increased. From statistical analysis, the average cell size displayed insignificantly different between that of the NRL foam at BLF loading at (0.00 and 2.50 phr) and (7.50 and 10.00 phr). The addition of BLF seemed to cause the foam cells to coalesce, resulting in fewer cell counts and larger pore sizes. Similar results were reported by Tomyangkul, S. et al. [33] and Surya, I. et al. [51]. It is commonly known that NRL foam structure dictates most foam properties. The foam structure could be tailored through various parameters, including foaming agent type and amount [18], fabrication methods, filler type, filler or particle size [32], filler loading [14], etc.

#### 3.1.4. Hardness

Hardness is the degree of firmness of a material—it is the ability to resist deformation, scratching, or abrasion [52]. The Shore OO type of durometer hardness was used in this research following ASTM D2240 due to the cushion foam as extremely soft rubber, sponges, and foams [43]. Cushion material with too much hardness can cause damage to the soft surface of a fruit. The hardness of the cushion foam samples is shown in Table 2. It could be seen that the hardness of EPE-FN and NRL-FN with and without BLF were similar. The result indicated that the hardness quality of the eco-friendly NRL-FN cushion could be used as an alternative to the EPE-FN fruit cushion. A slight increase in hardness was observed as the BLF content increased, but it was statistically insignificant. The finding was unexpected as other researchers had reported increases in hardness when fillers were added to NRL foams. Bashir, A. et al. added eggshell powder (ESP) to the NRL foam. The obtained NRL foam composite possessed improved harness and stiffness when the ESP content was at least 10 phr. The improvements were due to the EPS demobilizing the polymer chains of the NRL foam matrix [53]. Mahathaninwong, N. et al. also reported a similar finding of the increased hardness of NRL foam due to the incorporation of Agarwood-waste (ACW) powder [17]. They also suggested that the change in hardness was due to the change in foam cell morphology, contributing to the filler addition. Sirikulchaikij, S. et al. also reported the fabrication of NRL foam using a bubbling process. The hardness of the latex increased when the cell size became larger. Thus, it could be inferred that the hardness depended on the NRL foam’s cell structure or morphology [24]. In our work, as shown in Table 2 and Figure 8, adding BLF up to 10 phr slightly increased the cell size while the cell wall thickness values remained relatively constant. Therefore, the small change in hardness of the NRL-FN-BLF composites was in good agreement with the insignificant change in the foam morphology.

#### 3.1.5. Compression Set

The elastic behavior of elastomeric material is generally determined using a compression set test. A material with a low compression set can restore its thickness close to its original size after removing the compression force, suggesting a high material elasticity. On the other hand, a material with a high compression set may permanently lose its shape due to low elasticity. In general, a smaller permanent compression set leads to higher foam recoverability [54]. Figure 10 shows the compression set percentage of NRL foams with different BLF loadings. NRL-FN without BLF had a compression set of 12.5%. The addition of 2.5 phr BLF to NRL-FN insignificantly altered the compression set of the foam. The compression set percentage showed an increasing trend as a function of the BLF loading, with a maximum of 16.0% at 10 phr of BLF. A similar trend was observed when rice husk powder (RHP) was added to the NRL foam. The RHP was used to decrease the recovery percentage of the NRL foam. At low RHP loading, the NRL foam composite could be returned to its initial thickness faster than the foam with higher RHP loading [31]. Another report showed that better dispersion of a filler (kenaf) and smaller particle size led to a smaller deformation. The filler agglomeration decreased the elasticity behavior of the NRL foam. The agglomerated filler restricted the molecular chains‘ movements, enhancing the foam’s stiffness [51]. Moreover, foam with larger cells with thinner cell walls might be easily collapsed under the load. Thus, NRL foam with smaller cells and more cell walls, such as what was found in our samples, should be able to withstand compression deformation with good recoverability [17,54]. It should be noted that the compression set percentage reported here (maximum 16%) was significantly lower than those reported by the others (maximum 65%). This might be due to the closed-cell foam structure of our samples, which provided elasticity and resistance to compression deformation. For the target application of fresh produce’s cushion material, a lower compression set may be more desirable as the cushion foam with a high compression set may lose its ability to protect the packaged produce after being compressed.

#### 3.1.6. Compressive Test

The relationship between compressive stress and compressive strain of the EPE-FN and the NRL-FN with different BLF loadings is shown in Figure 11. Similar behavior was observed for both the commercial EPE-FN and NRL-FN samples, where the compressive stress gradually increased as the compressive strain increased. This behavior was common for a foam with a closed-cell structure. The compressive stress at 50% strain of EPE-FN was lower than those of NRL-FN at various BLF contents. The results suggested that the commercial EPE-FN was more readily deformed than the NRL-FN and NRL-FN-BLF composites. From the stress–strain curve in Figure 11, it could be seen that the NRL-FN and NRL-FN-BLF composites had larger areas under the curves than the EPE-FN. The larger areas under the curves of the eco-friendly foams indicated a greater ability to absorb energy over the EPE-FN commercial cushion materials.

The presence of BLF in NRL-FN slightly increased the compressive stress at 50% strain. An increasing trend of the compressive stress at 50% strain was observed as the BLF content in NRL-FN increased. The compressive stress at 50% strain of NRL-FN containing BLF contents of 0.00, 2.50, 5.00, 7.50, and 10.00 phr were 56.58, 58.89, 48.09, 81.14, and 77.85 kPa, respectively. The maximum stress was obtained when BLF 7.5 phr was added to NRL-FN. Further increased the BLF content to 10.0 phr showed insignificant change in the compressive stress at 50% compressive strain. The change in the compressive stress of a porous material with a filler depends on the type and content of the filler used as well as its effects on the foam cell structure. Typically, a foam’s compression stress and stiffness increase with increasing filler loading. Consequently, the increase in stiffness leads to an increment in hardness [31,36,55]. From our results, adding BLF insignificantly changed the foam morphology; the slight increase in the compressive stress at 50% strain possibly came from the presence of the harder particle of BLF.

In the packaging of fresh produce, the cushion material must absorb energy from external forces during transportation. Consequently, the lowest amount of energy may transfer to the packed fresh produce, thus preventing damage from occurring. The cushion that absorbs more energy is, therefore, better at reducing the bruising damage of fresh produce. The ability of a material to absorb energy could be determined in terms of “cushion coefficient (C)”. The cushion coefficient is determined from the compressive stress–strain curve, and the results are shown in Figure 12. It could be seen that the cushion coefficient was dependent on the strain percentage. A significant decline in the cushion coefficient was observed initially up to 45% strain before it leveled off. To compare the cushion coefficient of the samples, cushion coefficients at 50% compressive strain were chosen and shown in Table 3. It could be seen that the commercial EPE-FN cushion possessed the highest cushion coefficient of 5.24, indicating it had the lowest ability to absorb energy among all samples.

On the other hand, the cushion NRL-FN without BLF possessed the lowest cushion coefficient of 4.62. The presence of BLF and its increasing content in the NRL-FN cushion gave rise to the cushion coefficient. The results suggested that NRL-FN was the best material for the cushioning application as it could absorb the highest energy from the external force, which, in turn, was the most successful in minimizing the energy that otherwise might damage the packed fruit. A small amount of BLF used in the NRL-FN might be beneficial for enhancing the biodegradability of the cushion foam net without sacrificing the cushion performance. A pack test using the eco-friendly cushions with the actual model fruit is required to prove their effectiveness as a cushion material. Based on the available data, the encouraging results implied that the NRL-FN-BLFs might be used instead of EPE-FN for commercial purposes in the future.

### 3.2. Biodegradation Study

In the biodegradation study, the cushion foams of EPE-FN, NRL-FN, and NRL-FN-BLFs were buried in planting soil and placed outdoors for 24 weeks (May–October 2022). The biodegradation evaluations include the visual inspection of the appearance and their microstructure (SEM), weight loss percentage, chemical structure, and mechanical properties.

#### 3.2.1. Appearance

Table 4 shows the appearance (photographs on the left column) and surface microstructure (SEM micrographs on the right column) of the cushion samples before and after soil burial for 24 weeks. Before the soil burial experiment (week 0), the EPE-FN was white and cumulus. The NRL-FN was off-white with a yellowish tint. A greenish color was observed when BLF was incorporated into the NRL-FN, with the intensity of the green color increased as a function of the increasing BLF content. After 24 weeks of the soil burial test, the EPE-FN became slightly denser, and its color remained white. The NRL-FN and NRL-FN-BLFs, on the other hand, did not show a change in shape, but the color turned slightly brownish. The darker shade of the brownish color was more prominent as the BLF content in the NRL-FN-BLFs increased. The change in color was used as an indicator for the biodegradation of natural fiber/polymer composites. Luthra, P., Vimal, K.K., Goel, V. et al. [56] suggested that the color appearance was an important parameter used in the assessing the deterioration of the PP/natural fiber composite. The color change was mainly due to the changes in the chemical structure of the lignocellulosic complex of the natural fiber in the composite. A similar result was also reported by Butylina et al. [57]. It was also reported that removing the lignin from the fiber decreased the change in color. In our study, BLF was used without any treatment. Therefore, it could be inferred that the brownish color that appeared on the NRL-FN and NRL-FN-BLFs was a sign of degradation. The greater color changes of the NRL-FN-BLF cushion foams as the BLF content increased were in good agreement with the changes on the cushions’ surfaces observed in the SEM micrographs.

The SEM micrographs of the cushions’ foam surfaces showed significant changes in the surface appearance and roughness. The increases in surface roughness coupled with the emergence of microvoids became more pronounced as the BLF content increased. Similar results were reported by Shah, A.A., Hasan, F., Shah, Z. et al. [58]. They showed that a NR latex glove appeared rough with irregular cracks and pits after 2 weeks of exposure to *Bacillus sp. AF–666*. The greater surface erosion suggested higher degradation of NRL-FN with higher BLF contents. Similar increasing trends of degradation as a function of the increasing natural fiber contents were reported by others [56,57]. To confirm the degradation of the NRF-FN-BLF cushion foam, physical properties, weight loss, and chemical structure were investigated.

#### 3.2.2. Weight Loss

The weight loss values of the EPE -FN, NRL-FN, and NRL-FN-BLFs over 24 weeks of soil burial are shown in Figure 13. The weight loss percentage is a crucial indicator of the degradation of the samples. From the graph, it could be seen that EPE-FN showed a slight weight loss initially in the first 4 weeks before the weight loss became constant. The maximum weight loss was approximately 1%. The NRL-FN showed a greater degree of degradation than the EPE-FN. The fast-increasing weight loss was observed in the first 8 weeks before it leveled off. The maximum weight loss at 24 weeks of soil burial was about 4%. With the presence of BLF in NRL-FN, a greater initial weight loss was observed, and the weight loss continued to increase after week 8. It was noticeable that higher weight loss was observed with higher BLF content in NRL-FN. The degradation rate could be estimated from the slope of the curves in Figure 13. The results suggested a fast degradation rate was observed in the initial soil burial time before week 8. After week 8, the NRL-FN samples containing BLF continued to degrade with slower degradation rates. The rate tended to increase with the BLF content. At the end of soil burial at week 24, the weight loss of NRL-FN containing 10 phr of BLF was 1.8 times greater than that of NRL-FN without BLF. The results suggested that the natural fiber acted as a biodegradation accelerator for the NRL foam composites. The weight reduction was due to the microorganisms’ activity during soil burial, which took place on the skin of the samples and appeared as surface erosion and microvoids in the SEM micrographs in Table 4.

Tsuchii reported that a surgical glove was degraded in 20 days in a culture medium. This was probably because their samples were very thin compared to ours, and their controlled environment also played a crucial role in degrading their NR products. Soil burial was used in this study to simulate the normal ecological condition of the landfill method, the most common technique of waste management. The biodegradation rate may be accelerated significantly in a controlled environment with the right microbe (fungi and/or bacteria) strain [59]. Moreover, several reports suggested that some chemicals in the NR compound may enhance or prohibit the biodegradation of rubber products [60]. The use of BLF as a filler in the NRL-FN enhanced biodegradation of the eco-friendly foam cushion. Optimization of the chemicals used in the making of the NRL-F may be done to further increase the biodegradation rate.

#### 3.2.3. Fourier Transform Infrared (FTIR) Analysis

FTIR analysis method is a technique for observing the chemical structure of materials. FTIR spectra of the cushion foam samples before and after 24 weeks of soil burial at various BLF loadings (0.00, 2.50, 5.00, 7.50, and 10.00 phr) are compared, as shown in Figure 14. FTIR bands assignment of NRL, BLF, and NRL composites are shown in Table 5.

FTIR spectra of NRL-FN-BLF0.0 before soil burial showed peaks at 2800–3000 cm^−1^ (−CH stretching), 1655–1665 cm^−1^ (−C=C-stretching), and 835 cm^−1^ (=CH bending in isoprene backbone of NR). When compared to NRL foam, NRL foam composites exhibited absorption peaks at 3326 (−OH stretching vibration of cellulose, hemicellulose, and lignin), 2918 (C-H stretching of cellulose, hemicellulose, and lignin), 1628 (asymmetric stretching band of the carboxyl group of glucuronic acid in hemicellulose, −C=O stretching in conjugated of a carboxyl group), 1512 (−C=C-C-aromatic ring stretching and vibration in lignin), and 1033 (−C-O-stretching of cellulose and lignin) cm^−1^. The absorption peaks of BLF are shown in Figure 15.

The degradation of the vulcanized natural rubber is caused by direct action from microbes, which results in polymer chain cleavage and affects the functional group of the material [64]. The cleavage of the poly (cis-1,4-isoprene) backbone occurred first at the double bond via oxidative degradation [65]. After the soil burial, the band intensity of −OH stretching at 3326 cm^−1^ and −C=O stretching at 1739 cm^−1^ of NRL foam slightly increased, as shown in Figure 14. The formation of carbonyl and hydroxyl bonds (−C=O and –OH) was due to the oxidative degradation of NR [59]. In addition, the intensity of the band at 1020 cm^−1^ assigned to −C-O-C-stretching was marginally increased [62].

Moreover, the intensity of the peaks at 1628, 1541, and 1270 cm^−1^ (−C=O stretching of hemicellulose and lignin) and 1503 cm^−1^ (−C=C-C-aromatic ring stretching and vibration of lignin) decreased after soil burial due to the degradation of BLF. With increasing BLF content in the NRL foam composites, the increment of the intensity of −OH stretching, −C=O stretching, and −C-O-C-stretching were seen in Figure 16. This may be because BLF improved the degradation of the composites. The biodegradability of the NRL foam composites was confirmed by the weight loss percentage and morphology.

#### 3.2.4. Compressive Test

Generally, a decline in physical properties is expected as NR products are degraded [64]. The 50% compressive strength of cushion foams at over 24 weeks of soil burial time is shown in Figure 17. The 50% compressive strength of EPE-FN was unchanged over the 24-week period. The compressive strength of the NRL-FN without and with BLF showed a decreasing trend in the first 12 weeks but became constant or even increased (such as in the case of 10 phr of BLF in NRL-FN). The slight decreases in compressive strength in the NRL-FN samples were expected as the biodegradation damaged the samples’ structure and integrity. The loss in long hydrocarbon chains also weakens the rubber properties. The increase in 50% compressive strength in some samples in the later weeks was plausibly because the degraded cushion became stiffer, hence more resistance to compressive force. Overall, the degradation of the foam sample took place, and the BLF increased the degradation rate of the NRL-FN. However, the changes in properties were relatively low. This result may be beneficial as it suggests that the eco-friendly NRL cushion could be continuously used or reused for a prolonged period of time. A well-designed logistical strategy must be created for recovering/returning the cushion after use.

## 4. Conclusions

The eco-friendly cushions of NRL foam were developed as an alternative for cushion packaging for fresh produce. The NRL was a primary material from a renewable resource. The Dunlop process and microwave irradiation used in the vulcanization step were energy efficient. The addition of BLF to the NRL-FN improved the cushion biodegradability. The presence of BLF also affected other properties of the NRL foams, including processability, bulk density, and foam cell structure. Thus, the mechanical properties and cushion coefficient were also affected.

This study illustrated the possibility of improving the packaging cushion to be more sustainable and lessen its negative environmental impact. The presence of BLF positively enhanced the biodegradation of the cushion up to 1.8 times (weight loss) compared to the NRL-FN without BLF in 24 weeks. Meanwhile, the EPE commercial cushion foam was indifferent in weight during the same period under the same ecological conditions. Other properties, including bulk density, compression set, compressive strength and cushion coefficient, were slightly increased but statistically insignificant. To verify the cushioning performance of the eco-friendly NRL foam, the pack test of the foam with a model fruit (guava) has been carried out. The results, reported elsewhere [66], are encouraging and provide insight into further improving the cushioning packaging.

In this study, 10 phr of BLF added to the NRL-FN did not significantly alter the cell foam size and structure. The results suggested that the process used in creating the eco-cushion foam net was very forgiving. One might add other different fillers or additives to the cushion foam to create extra functions, such as acting as an ethylene absorber and antibacterial and/or anti-browning agents. Moreover, other natural fibers can also be used to enhance the biodegradability of the NRL cushion foam, including rice husk, banana, hemp, pineapple, bamboo, coconut, etc. Hence, the natural fibers from agricultural waste may become valuable material. Ultimately, an alternative cushion that is both smart and eco-friendly may be obtainable in the near future.

## Figures and Tables

**Figure 1 polymers-15-00654-f001:**
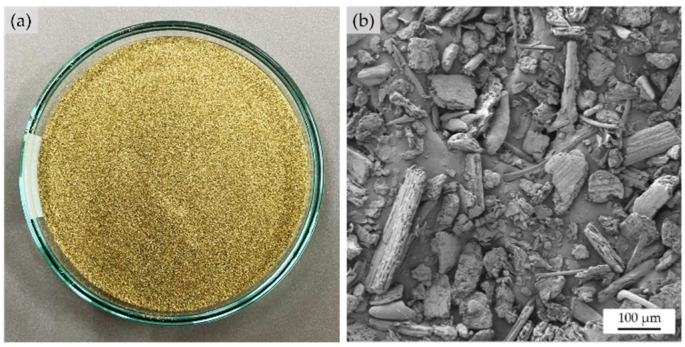
The appearance and shape of bamboo leaf fibers: (**a**) photograph; (**b**) SEM micrograph at a 100× magnification.

**Figure 2 polymers-15-00654-f002:**
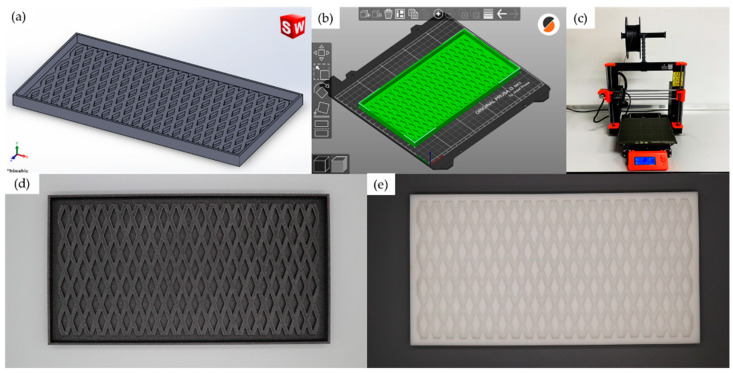
Schematic for fabricating a rapid silicone mold using 3D printing: (**a**) 3D model; (**b**) converting to G-code; (**c**) 3D printing; (**d**) inverted mold; and (**e**) silicone mold.

**Figure 3 polymers-15-00654-f003:**
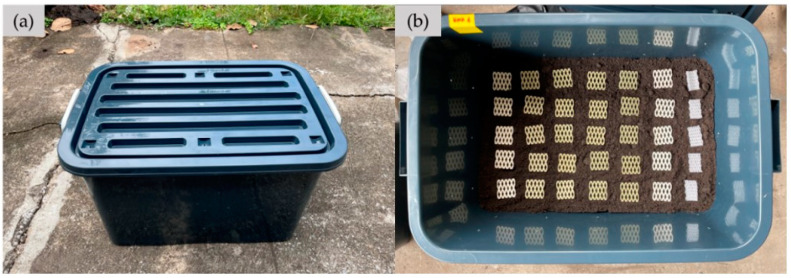
Biodegradation experiment of cushion foams: (**a**) plastic box used for soil burial; (**b**) cushion foam placement.

**Figure 4 polymers-15-00654-f004:**
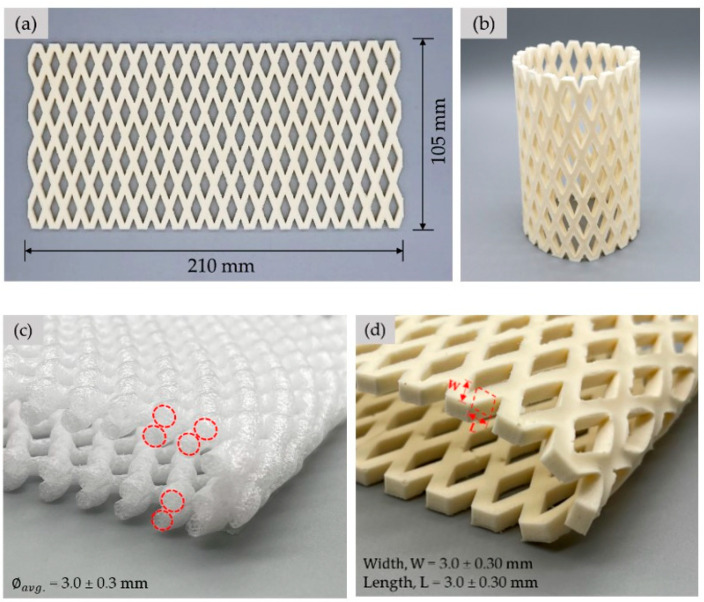
Fabrication and design of NRL-FN: (**a**) foam net sheet; (**b**) foam net cylindrical tube; (**c**) cross-section of EPE-FN; and (**d**) cross-section of NRL-FN.

**Figure 5 polymers-15-00654-f005:**
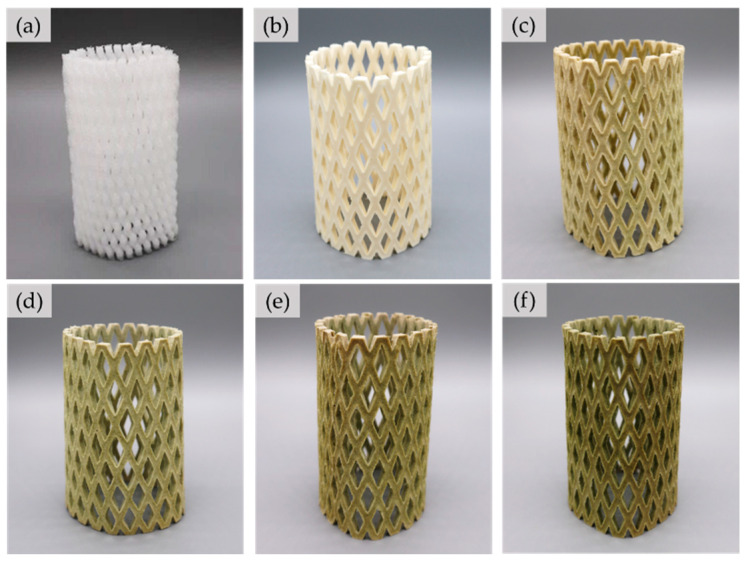
Cushion foam net: (**a**) EPE-FN; (**b**) NRL-FN-BLF0.0; (**c**) NRL-FN-BLF2.5; (**d**) NRL-FN-BLF5.0; (**e**) NRL-FN-BLF7.5; (**f**) NRL-FN-BLF10.0.

**Figure 6 polymers-15-00654-f006:**
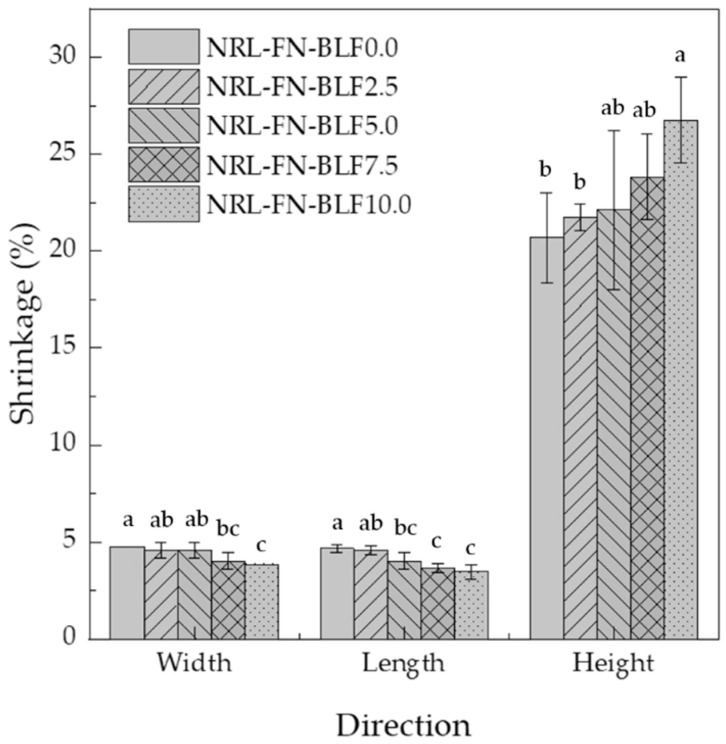
Shrinkage percentage of NRL foam composite at various BLF loading at different shrinkage directions compared with silicone mold size. Significant differences at *p* < 0.05 are shown by different letters. The values represent the mean ± S.E. of five replicates.

**Figure 7 polymers-15-00654-f007:**
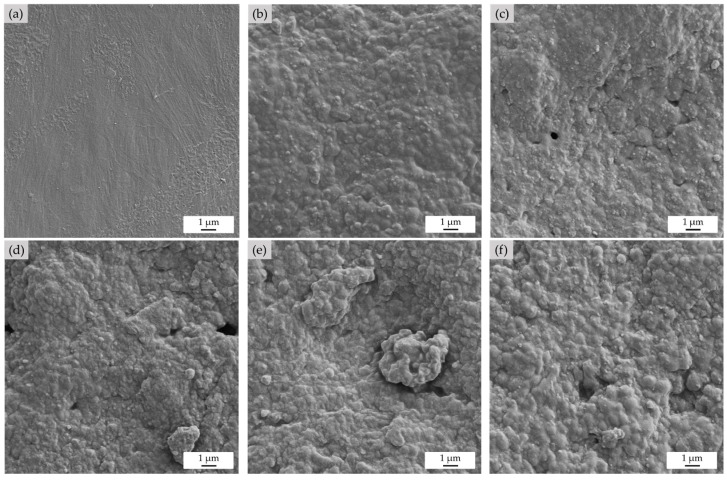
NRL foam composites morphology at the top surface of specimens (5k× magnification): (**a**) EPE-FN; (**b**) NRL-FN-BLF0.0; (**c**) NRL-FN-BLF2.5; (**d**) NRL-FN-BLF5.0; (**e**) NRL-FN-BLF7.5; (**f**) NRL-FN-BLF10.0.

**Figure 8 polymers-15-00654-f008:**
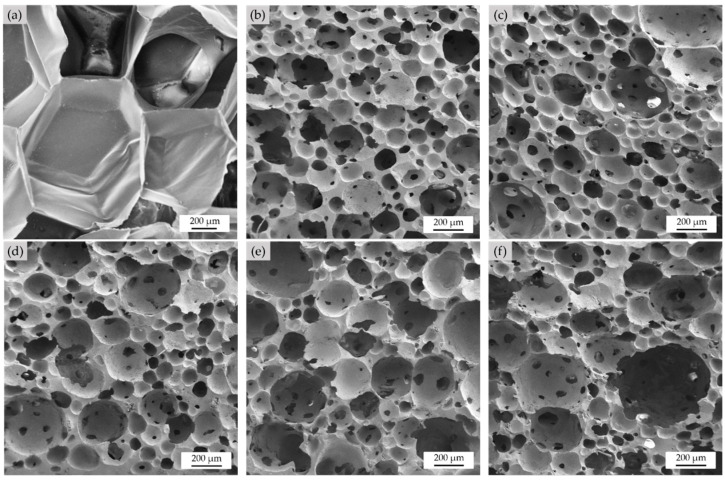
Cushion foams morphology at a 50× magnification: (**a**) EPE-FN; (**b**) NRL-FN-BLF0.0; (**c**) NRL-FN-BLF2.5; (**d**) NRL-FN-BLF5.0; (**e**) NRL-FN-BLF7.5; (**f**) NRL-FN-BLF10.0.

**Figure 9 polymers-15-00654-f009:**
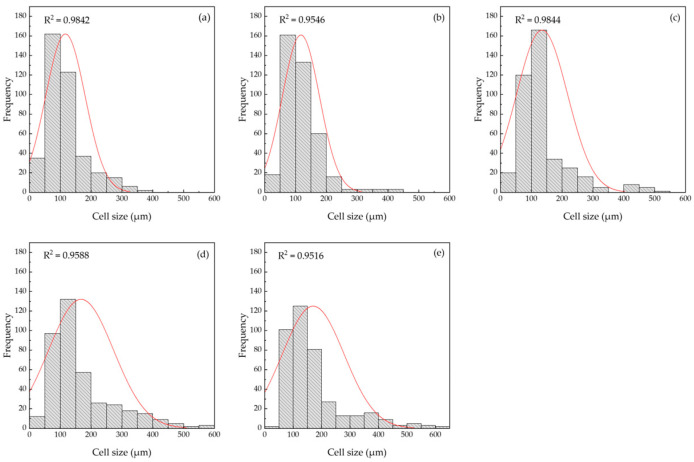
NRL foam composites distribution curve with various BLF loadings (*p* < 0.05, *n* = 400): (**a**) NRL-FN-BLF0.0; (**b**) NRL-FN-BLF2.5; (**c**) NRL-FN-BLF5.0; (**d**) NRL-FN-BLF7.5; (**e**) NRL-FN-BLF10.0.

**Figure 10 polymers-15-00654-f010:**
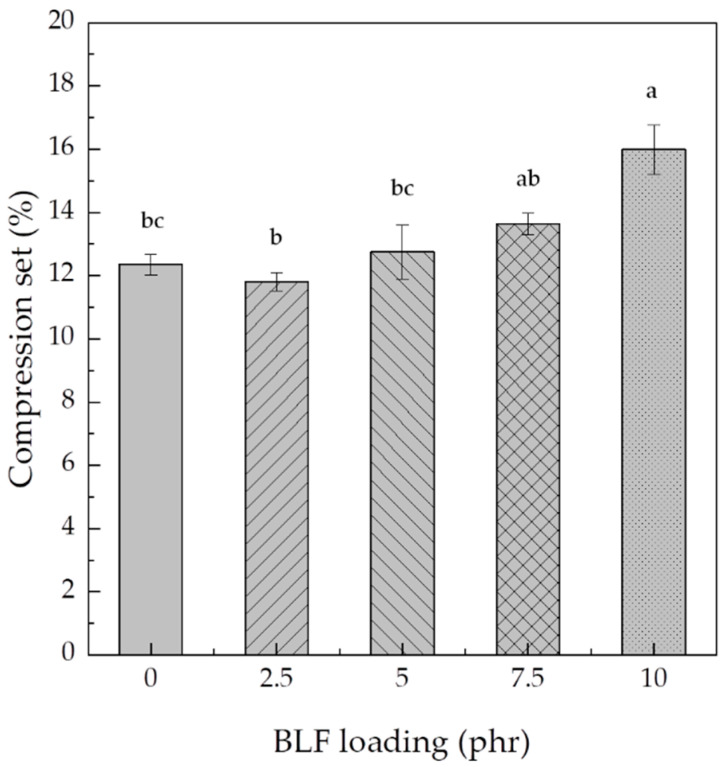
Compression set of NRL foam composite various BLF loading. Significant differences at *p* < 0.05 are shown by different letters. The values represent the mean ± S.E. of five replicates.

**Figure 11 polymers-15-00654-f011:**
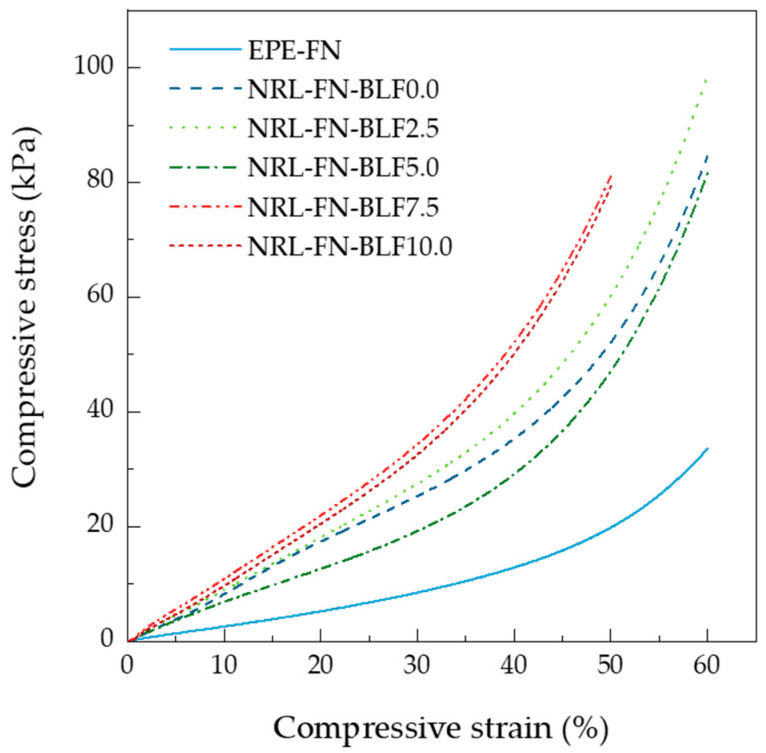
Stress–strain curve of the cushion foams.

**Figure 12 polymers-15-00654-f012:**
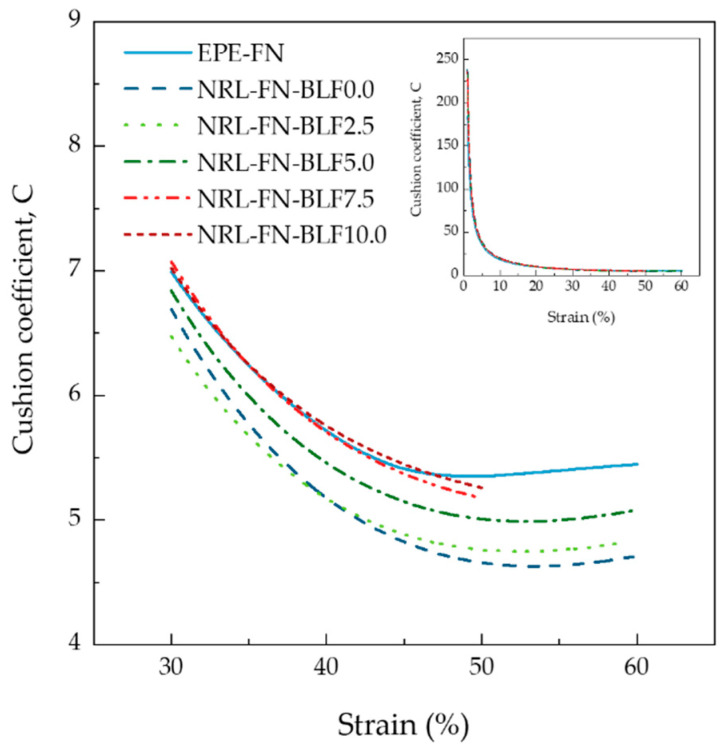
Cushion coefficient–compressive strain curves of cushion foams.

**Figure 13 polymers-15-00654-f013:**
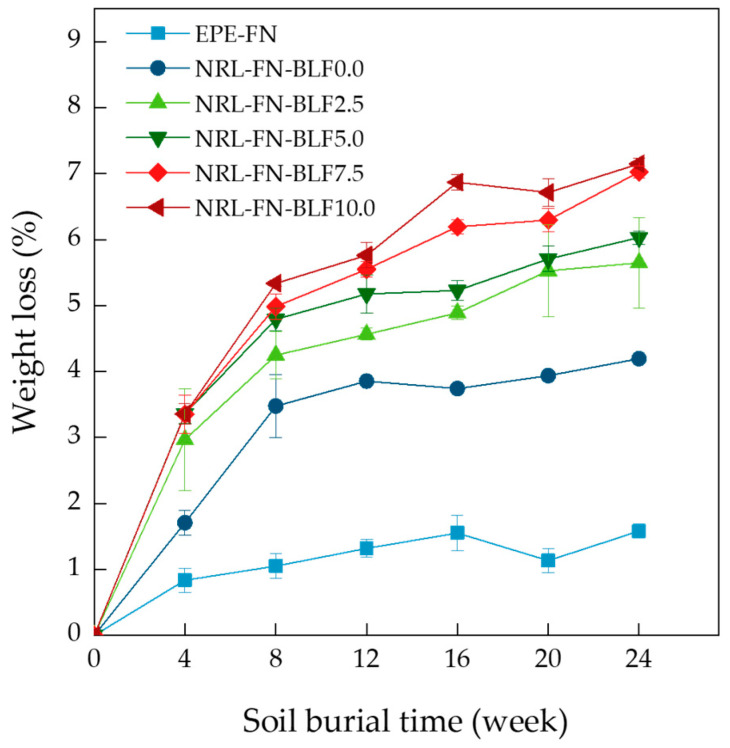
Weight loss percentage of EPE-FN, NRL-FN, and NRL-FN-BLFs cushion foams at various BLF contents over 24 weeks of soil burial times.

**Figure 14 polymers-15-00654-f014:**
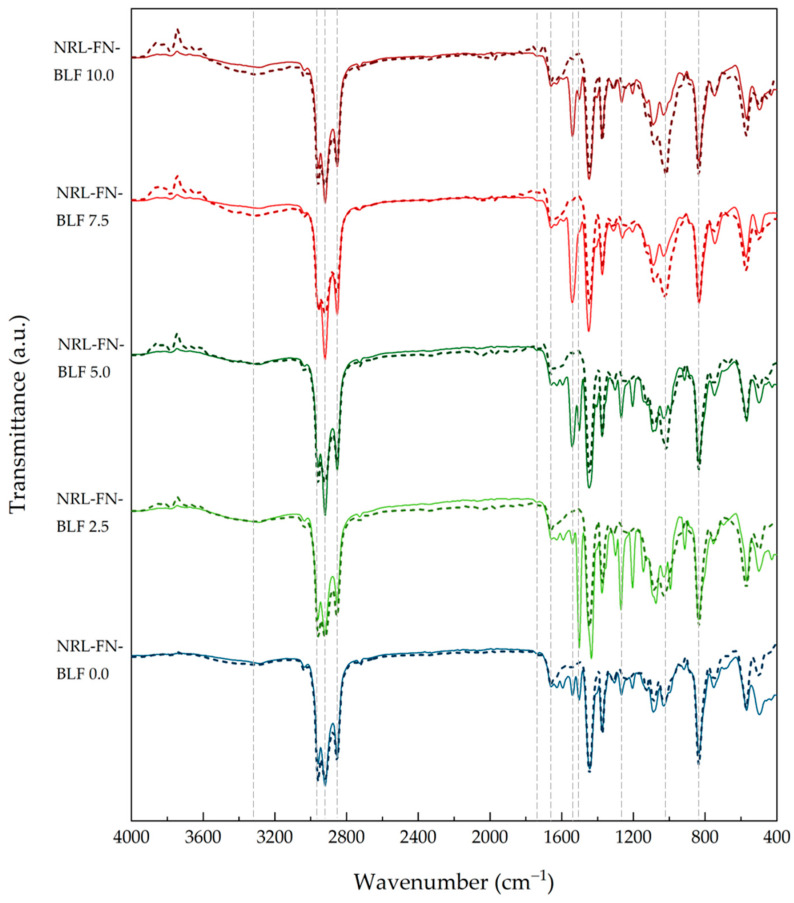
FTIR spectra of NRL foam composites before (solid lines) and after (dash lines) soil burial.

**Figure 15 polymers-15-00654-f015:**
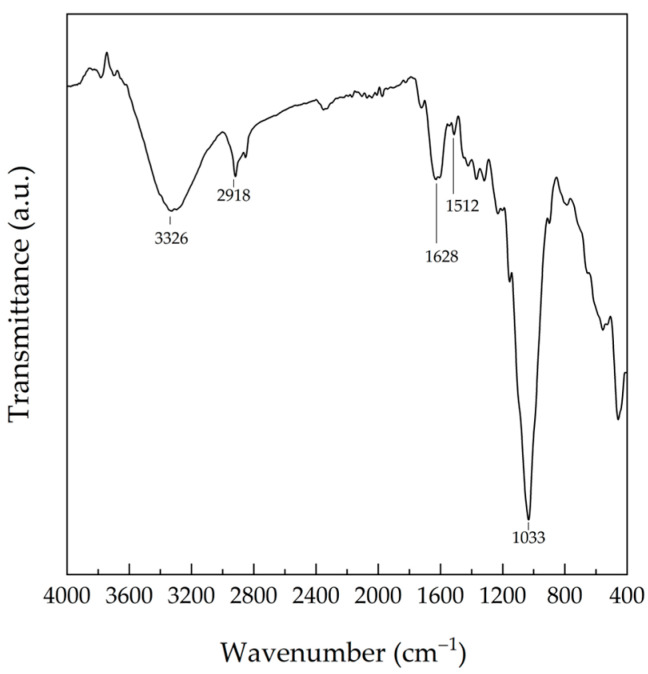
FTIR spectra of BLF.

**Figure 16 polymers-15-00654-f016:**
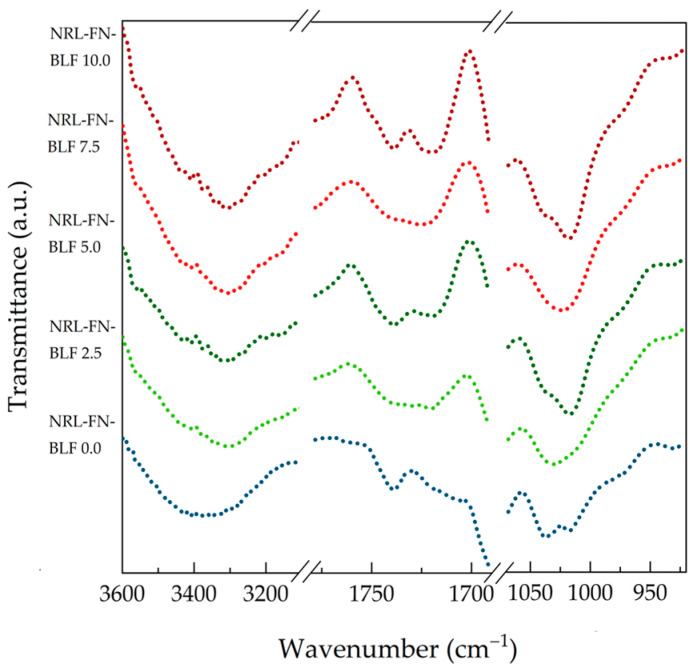
FTIR spectra of NRL foam composite at various BLF contents after soil burial for 24 weeks.

**Figure 17 polymers-15-00654-f017:**
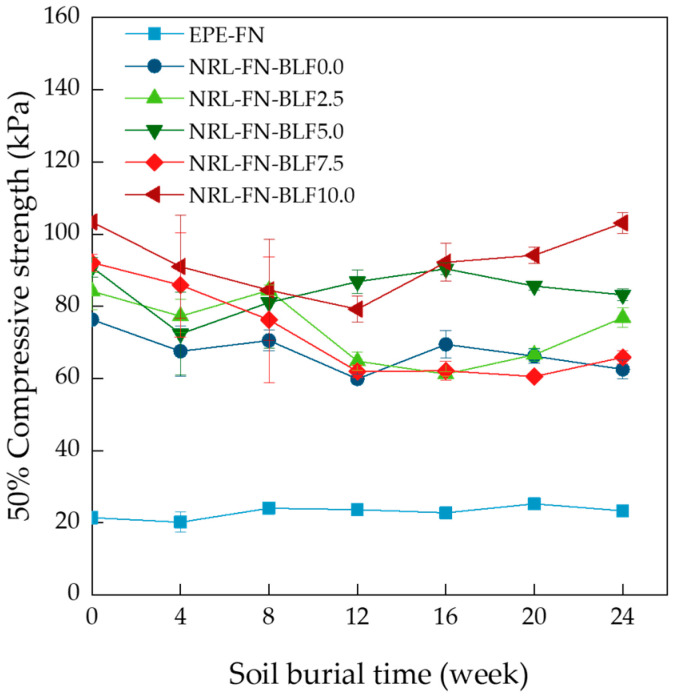
Compressive strength of 50% of cushion foam at various soil burial times.

**Table 1 polymers-15-00654-t001:** Formulation of NRL foam composite at various BLF loadings.

Ingredients	Content (phr ^1^)	Functions
60% High ammonia natural rubber latex (HA Latex)	100.00	Matrix
10% Potassium oleate (K-oleate)	4.50	Foaming agent
50% Sulfur	2.00	Vulcanizing agent
50% Zinc diethylthiocarbonate (ZDEC)	2.00	1st accelerator
50% Zinc 2-mercaptobenzothiazone (ZDMT)	2.00	2nd accelerator
50% Wingstay L	2.00	Antioxidant
50% Zinc oxide (ZnO)	5.00	Activator
12.5% Sodium silicofluoride (SSF)	1.00	1st gelling agent
33% Diphenyl guanidine (DPG)	1.40	2nd gelling agent
Bamboo leaf fiber (90 ≤ x ≤ 106 µm)	0.00, 2.50, 5.00, 7.50, 10.00	Natural fiber

^1^ Parts per hundred rubber.

**Table 2 polymers-15-00654-t002:** Foam properties of EPE-FN and NRL-FN-BLFs.

Cushion Foams	Bulk Density (kg/m^3^)	Hardness (Shore OO)	Number of Cells per Unit Volume (Cells/cm^3^)	Average Cell Size (µm)
EPE-FN	18.869 [44]	29.56 ± 0.65 ^a^	-	-
NRL-FN-BLF0.0	264.59 ± 4.64 ^b^	28.88 ± 0.61 ^ab^	63,726.67	93.54 ± 2.64 ^c^
NRL-FN-BLF2.5	272.56 ± 5.35 ^ab^	28.64 ± 0.67 ^b^	56,892.56	100.73 ± 4.76 ^c^
NRL-FN-BLF5.0	276.99 ± 7.22 ^ab^	29.08 ± 0.78 ^ab^	50,968.49	108.06 ± 2.06 ^b^
NRL-FN-BLF7.5	281.53 ± 11.60 ^a^	29.28 ± 0.67 ^ab^	51,381.86	115.05 ± 3.66 ^a^
NRL-FN-BLF10.0	288.47 ± 12.51 ^a^	29.48 ± 0.48 ^b^	45,758.51	122.58 ± 4.52 ^a^

Note: Significant differences at *p* < 0.05 are shown by different letters. The values represent the mean ± S.E. of five replicates.

**Table 3 polymers-15-00654-t003:** Cushion coefficient at 50% strain of a commercial cushion foam compared to the eco-friendly cushion NRL-FN at various BLF contents.

Cushion Foams	Cushion Coefficient (c) at 50% Strain
EPE-FN	5.24
NRL-FN-BLF0.0	4.62
NRL-FN-BLF2.5	4.81
NRL-FN-BLF5.0	5.01
NRL-FN-BLF7.5	5.19
NRL-FN-BLF10.0	5.21

**Table 4 polymers-15-00654-t004:** Appearance and micrograph of cushion foams before soil burial (0 weeks) and after soil burial (24 weeks).

Cushion Foams	Foams Appearance		Foams Micrograph	
	Before Soil Burial	After Soil Burial	Before Soil Burial	After Soil Burial
EPE-FN	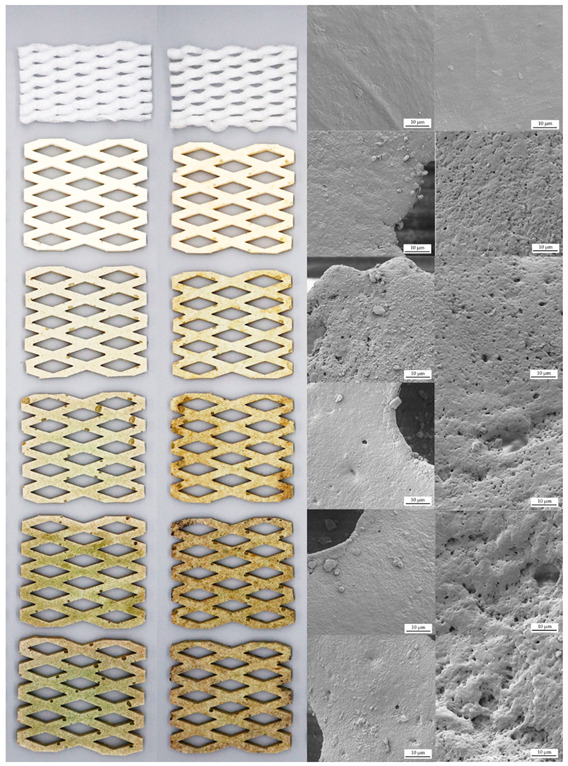			
NRL-FN-BLF0.0				
NRL-FN-BLF2.5				
NRL-FN-BLF5.0				
NRL-FN-BLF7.5				
NRL-FN-BLF10.0				

**Table 5 polymers-15-00654-t005:** IR bands assignment of NRL, BLF, and NRL composite.

Wavenumber (cm^−1^)	Assignment
**3300–3380**	−OH stretching vibration (−OH as a result of the degradation by oxidation) [59]
**2958–2960**	−CH_3_ asymmetric stretching vibration of natural rubber [59]
**2919–2927**	−CH_2_ asymmetric stretching vibration of natural rubber [59]
**2852–2854**	−CH_2_ asymmetric stretching vibration of natural rubber [59]
**1710–1740**	Carbonyl group (C=O) from ketone or aldehyde results from oxidative degradation [59]
**1655–1665**	−C=C-stretching vibration in the NR structure or maybe due to absorbed water or carboxylate or conjugated ketone (−C=O) resulted from the degradation [59]
**1618**	Aromatic skeletal vibration, C=O stretching, absorbed O-H of hemicellulose and lignin [60] of BLF
**1508**	C=C-C aromatic ring stretching and vibration of lignin [60] of BLF
**1537, 1270**	C=O and –O-R of hemicellulose [61] of BLF
**1032**	C-O stretching, aromatic C-H in-plane deformation of cellulose and lignin [60] of BLF
**1020**	−C-O-C-stretching of esther group [62]
**870–830**	Isoprene backbone of NR [63]

## Data Availability

The data presented in this study are available upon request from the corresponding author.

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
