# Peer review of "Effects of Bamboo Leaf Fiber Content on Cushion Performance and Biodegradability of Natural Rubber Latex Foam Composites"

_polymers, 2023, doi:10.3390/polym15030654_

Round 1

Reviewer 1 Report

The manuscript proposes the study of the use of bamboo leaf fibers in vulcanized natural rubber foams.

The problem is well described and the quality and theoretical and experimental approaches used are adequate for the problem.

The critical point, in my opinion, is in the description of the vulcanization of the natural rubber compound. It was not detailed how the microwave curing conditions were achieved. The cited article that should contain this information (Ref. 17) does not address microwave vulcanization. In addition to this problem, batch heating in conventional ovens generates hot spots that can cause differences in the degree of vulcanization in different parts of the piece. Was any temperature measurement taken before or after the vulcanization step at different points on the piece? Were the properties of different regions of the vulcanized part evaluated? This step deserves greater care, as it will have a huge influence on the measured properties.

Apart from this point, I consider that the article has great merit and quality for publication.

Author Response

Dear Editor and Reviewers,

Thank you for giving us the opportunity to submit our revised manuscript entitled "Effects of bamboo leaf fiber content on cushion performance and biodegradability of natural rubber latex foam composites," to the Polymers. We appreciate the time and effort you and the reviewers took to offer your insightful comments on our work. We have revised our manuscript using the track changes function in the MS Words. The attached file is the point-by-point responses to the reviewers’ comments and concerns.

We hope that the revised manuscript has met the reviewers’ requirements.

Best Regards,

Tatiya Trongsatitkul, Ph.D.

Assistant Professor

School of Polymer Engineering

Institute of Engineering

Suranaree University of Technology

[email protected]

Reviewer 2 Report

The manuscript entitled “Effects of bamboo leaf fiber content on cushion performance and biodegradability of natural rubber latex foam composites” has been reviewed. The results are helpful. However, the manuscript needs to be well improved before the acceptance. Detailed comments are as follows:

1.     The full names of the abbreviations, such as HDPE, PP, LDPE, SEM, DI should be provided where they are first mentioned in the main text.

2.     The abbreviations, such as FTIR, NRL foam, BLF and GO, should be named only once in the whole main text.

3.     The full names of EPS, EPE and PLGA are expanded polystyrene foam, expanded polyethylene foam and poly(lactic-co-glycolic acid)s, not expanded polystyrene foams, expanded polyethylene foams and poly(lactic-co-glycolic acid).

4.     Correct the unnecessary capitalization of first letters in some phrases, such as Poly(3-hydroxybutyrate-co-3-hydroxyvalerate), Universal Testing Machine.

5.     In Line 230-231, the statements should be in one paragraph.

6.     Remove % from the phrases, such as Compression set (%) and Mold shrinkage (%).

7.     Please add the explanation for a, b and c in the figure captions in Fig. 6 and 10 and footnotes of Table 2.

8.     In Table 2, what does OO represent for?

9.     Please provide the unit of cushion coefficient.

10.  In the caption of Fig. 12, Cushion coefficient compressive strain should be Cushion coefficient-compressive strain.

11.  In Figs. 14 and 16, % should be a. u.

12.  In Fig. 14, please make the spectra before and after soil burial.

13.  In References, please unify the capitalization of first letters (e.g. Ref. 53) and journal names (e.g. Ref. 60). Refs. 64 and 65 are the same. Please check the references piece by piece.

Author Response

(The authors gave the same response as above.)

Round 2

Reviewer 2 Report

The manuscript has been well revised. It can be accepted if the following comments are considered:

1.     The abbreviations, such as FTIR, should be defined only once in the whole main text.

2.     In Fig. 15, a.u. should be %. Ticks and numbers should be included in the y-axis. In the caption, spectra should be spectrum.